# Use of A Hydroalcoholic Extract of *Moringa oleifera* Leaves for the Green Synthesis of Bismuth Nanoparticles and Evaluation of Their Anti-Microbial and Antioxidant Activities

**DOI:** 10.3390/ma13040876

**Published:** 2020-02-15

**Authors:** Prince Edwin Das, Amin F. Majdalawieh, Imad A. Abu-Yousef, Srinivasan Narasimhan, Palmiro Poltronieri

**Affiliations:** 1Asthagiri Herbal Research Foundation, 162A, Perungudi Industrial Estate, Perungudi, Chennai 600096, India; prince.ahrf@gmail.com; 2Department of Biology, Chemistry and Environmental Sciences, American University of Sharjah, Sharjah, P.O. Box 26666, UAE; iabuyousef@aus.edu; 3Institute of Sciences of Food Productions, CNR-ISPA, 73100 Lecce, Italy

**Keywords:** *Moringa oleifera*, bismuth nanoparticles, polyphenolics, anti-bacterial, anti-fungal, antioxidant

## Abstract

The employment of plant extracts in the synthesis of metal nanoparticles is a very attractive approach in the field of green synthesis. To benefit from the potential synergy between the biological activities of the *Moringa oleifera* and metallic bismuth, our study aimed to achieve a green synthesis of phytochemical encapsulated bismuth nanoparticles using a hydroalcoholic extract of *M. oleifera* leaves. The total phenolic content in the *M. oleifera* leaves extract used was 23.0 ± 0.3 mg gallic acid equivalent/g of dried *M. oleifera* leaves powder. The physical properties of the synthesized bismuth nanoparticles were characterized using UV-Vis spectrophotometer, FT-IR spectrometer, TEM, SEM, and XRD. The size of the synthesized bismuth nanoparticles is in the range of 40.4–57.8 nm with amorphous morphology. Using DPPH and phosphomolybdate assays, our findings revealed that the *M. oleifera* leaves extract and the synthesized bismuth nanoparticles possess antioxidant properties. Using resazurin microtiter assay, we also demonstrate that the *M. oleifera* leaves extract and the synthesized bismuth nanoparticles exert potent anti-bacterial activity against *Escherichia coli*, *Klebsiella pneumoniae*, *Staphylococcus aureus*, and *Enterococcus faecalis* (estimated MIC values for the extract: 500, 250, 250, and 250 µg/mL; estimated MIC values for the bismuth nanoparticles: 500, 500, 500, and 250 µg/mL, respectively). Similarly, the *M. oleifera* leaves extract and the synthesized bismuth nanoparticles display relatively stronger anti-fungal activity against *Aspergillus niger*, *Aspergillus flavus*, *Candida albicans*, and *Candida glabrata* (estimated MIC values for the extract: 62.5, 62.5, 125, and 250 µg/mL; estimated MIC values for the bismuth nanoparticles: 250, 250, 62.5, and 62.5 µg/mL, respectively). Thus, green synthesis of bismuth nanoparticles using *M. oleifera* leaves extract was successful, showing a positive antioxidant, anti-bacterial, and anti-fungal activity. Therefore, the synthesized bismuth nanoparticles can potentially be employed in the alleviation of symptoms associated with oxidative stress and in the topic treatment of Candida infections.

## 1. Introduction

*Moringa oleifera* (family Moringaceae) is among the most studied herbs. It is considered a sacred plant and referred to as the “tree of life” in several cultures. It has been widely cultivated in various parts of the world and for many decades because of its potential use in numerous applications in industry, pharmacy, and medicine. Both its leaves and fruits are used in cooking, and they are part of many cuisines worldwide. Moreover, many home remedies and traditional medicinal formulations are prepared using every part of *M. oleifera* [1]. Its leaves are edible and they are of considerable nutritional and therapeutic value due to their rich vitamin and amino acid content [2]. Experimental evidence suggests that the diverse medical applications of *M. oleifera* leaves can be partially attributed to the antioxidant potential of the phenolic compounds found in the leaves [3]. For many years, plant-based compounds and their derivatives have been used in the manufacturing of textiles, fabrics, polymers for food and non-food applications, biomaterials for biofilm formation as well as oral hygiene and dental caries prevention, and foil polymers for food packaging and wrapping [4,5,6,7,8,9]. The antimicrobial and antioxidant properties of silver and other metallic nanoparticles forced the development of several new nanomaterials, followed by the evaluation of their biological activities [10,11,12,13,14,15].

Several active compounds (e.g., proteins, flavonoids and carboxylic groups of arabinose and galactose, reducing sugars, aliphatic amines, tannins, aliphatic alkenes of alkaloids, aromatic amines, polysaccharides, sec-alcohols, saponins, and water-soluble heterocyclic components) isolated from plant extracts have been used in the successful synthesis of silver nanoparticles [16]. Individually, many plant extracts and metallic nanoparticles are known to possess significant biological activities both in vitro and in vivo. As such, it is envisioned that the use of plant extracts in the green synthesis of metallic nanoparticles may cause a synergistic effect leading to more robust biological activities. Hence, the use of plant extracts to synthesize metallic nanoparticles may lead to the formation of new nanomaterial with more potent and/or novel biological activities [17]. Indeed, several studies demonstrate the successful synthesis of silver and gold nanoparticles using the leaves’ extracts of *Erythrina suberosa (Roxb.)*, *Paederia foetida*, *Acalypha indica*, *Cassia auriculata*, *Sorbus aucuparia*, and *Azadirachta indica*, and their antimicrobial activities were assessed and confirmed [18,19,20,21,22,23]. Along the same line, other studies demonstrated that silver nanoparticles that were synthesized using various plant extracts exert potent antioxidant activity [24,25,26,27,28,29].

Bismuth nanoparticles with a size of 25 nm were successfully synthesized by using the laser ablation method, and they can be utilized as high contrast medium for high-resolution imaging in several biological applications [30]. Notably, bismuth nanoparticles synthesized by conventional solvent techniques have a size in the range of 50–103 nm with a very high anti-wear property [31]. Bismuth nanoparticles are also used as catalysts in reduction of nitro compounds to azo compounds [32]. Bismuth nanoparticles, with a size of 40 nm, which were synthesized by using the colloidal-chemical method in water medium, were shown to exert potent anti-microbial activity against several pathogenic microorganisms [33].

Bismuth nanoparticles with better catalytic activity were synthesized by the redox reactions between ammonium bismuth citrate and sodium borohydride in the presence of soluble starch in aqueous phase [34]. Bismuth nanodots were synthesized by the redox reaction between D-glucose and bismuth nitrate, in the presence of polyvinylpyrrolidone in the basic aqueous phase. The reduction of 4-nitrophenol to 4-aminophenol requires 36 µg/mL nano-catalyst for 20 mM of the substrate [35]. A mechano-chemistry pathway was used to prepare nanostructured Ag-Bi alloys under normal conditions and with the minimum amount of solvent. These nanostructured Ag-Bi alloys effectively degraded NO but they were rapidly oxidized [36]. Bismuth nanoparticles-decorated multi-wall carbon nano-tubes were synthesized and modified carbon paste electrodes were fabricated to serve as a novel amperometric sensor for the purpose of gallic acid detection. This sensor estimates gallic acid in cloves and green tea extracts with complete elimination of interferences [37]. Zerovalent bismuth nanoparticles were synthesized by a two-pot synthetic approach based on a combination of cementation process and a wet ball-stirring process [38]. Numerous studies were reported on the synthesis and characterization of bismuth sulfide, bismuth oxide, and spherical bismuth metallic nanoparticles. Various strategies and different capping agents were used to control the morphology, size, and stability of these nanoparticles. In the medical field, bismuth nanoparticles were used as an X-ray contrast agent [39]. Under a low X-ray dose, a radiosensitizer comprised of bismuth nanoparticles attached to antibodies was used to specifically and selectively target and kill multidrug resistant bacteria [39].

Herein, we attempted to achieve enhanced biological activities of bismuth nanoparticles via their synthesis using *M. oleifera* leaves’ phytochemical constituents. To this end, we pursued green synthesis of bismuth nanoparticles using a hydroalcoholic extract of *M. oleifera* leaves, followed by their physical characterization using different techniques. We also assessed the antioxidant activity of *M. oleifera* leaves extract and of synthesized phytochemical encapsulated bismuth nanoparticles, as well as their anti-bacterial and anti-fungal activities against different species of bacteria and fungi. We anticipate that our findings will be significant in the employment of green-synthesized bismuth nanoparticles in several medical applications.

## 2. Results and Discussion

### 2.1. Phytochemical Analysis of the M. oleifera Leaves Extract

The qualitative evaluation of various chemical constituents in the *M. oleifera* leaves extract was performed using the test methods mentioned in Table 1. The protein content in the *M. oleifera* leaves extract was estimated to be 0.1% of the dry leaves powder. The total phenolic content in *M. oleifera* leaves extract was 23% of the dry leaves powder. An earlier study demonstrated that alanine, tyrosine, lysine, and threonine are among the major amino acids present in the *M. oleifera* leaves extract [40]. The concoction of *M. oleifera* leaves extract contains isomers of ceffeoylquinic acid, isomers of feruloylquinic acid, tannins, gallic acid, and several flavonoids like quercetin, kaempferol (Appendix A) and their glycoside derivatives [41]. Using High-performance liquid chromatography (HPLC) analysis, isoquercetin, astragalin, and crypto-chlorogenic acid were identified in the ethanolic extract of *M. oleifera* leaves [42]. The hydroalcoholic extract of *M. oleifera* leaves (water/ethanol 50:50), contains numerous unexplored macromolecules. Collectively, such findings suggest that the *M. oleifera* leaves extract could serve as a nutritional supplement and a possible stabilizing agent for the synthesized bismuth nanoparticles.

From the total phenolic content estimation of the *M. oleifera* leaves extract before and after the reaction (Table 2), Scheme 1 is proposed. The total phenolic content of 60 mg gallic acid equivalent from 10 g of dried *M. oleifera* leaves powder was used in the synthesis of 170 mg of bismuth nanoparticles from 0.04 M of bismuth (III) ion solution. The reducing and binding chemical entities present in the concoction aid in the formation and stabilization of the synthesized bismuth nanoparticles through encapsulation. A color change from light brown to dark brown indicates the successful formation of bismuth nanoparticles.

### 2.2. Characterization

#### 2.2.1. Size and Morphology of the Synthesized Bismuth Nanoparticles

The high-resolution transmission electron microscopy (HRTEM) image of synthesized bismuth nanoparticles was captured using a JEOL-TEM 2100 plus electron microscope (Figure 1). EDS (energy dispersive X-ray spectroscopy) studies indicate only the presence of elements in the sample (Figure 2).

The synthesized encapsulated bismuth nanoparticles were washed only with ethanol and water, to avoid the use of harsh chemicals and thermal processes. Figure 1a shows the phytochemicals in the extract enclosing the nanoparticles as dark spots in the cloud of organic matrix. The right lower part of Figure 1a was enlarged to 20 nm in Figure 1b and further to 5 nm in Figure 1c. The selected area electron diffraction (SAED) of this portion was given in Figure 1d. The EDS, taken at the sample shown in Figure 1c, is given in Figure 2a,b.

Figure 2a,b presenting the counts at 2.469 KeV and 10.839 KeV correspond to the X-ray energies for bismuth Mα and Lα lines. The counts for bismuth come from the grid that holds the sample. The phytochemical matrix enclosing the sample explains the presence of other signals in the EDS. This indicates the presence of bismuth nanoparticles in the synthesized encapsulated nanoparticles and absence of other elements.

The synthesized bismuth nanoparticles were obtained in ethanol to give a colloidal solution. The electron diffraction pattern in Figure 1d explains the amorphous in nature of the product as nanoparticles agglomerate upon storage. EDS analysis (Figure 2) confirmed the presence of elemental bismuth nanoparticles. The absence of other elemental peaks reflects the purity of the sample. As shown in Figure 3a, the SEM images suggest an approximate size of the synthesized bismuth particles in the range of 40 and 60 nm. It is envisaged to perform a TEM analysis to obtain the proper size of nanoparticles.

However, SEM analysis suggests that phytochemical encapsulated bismuth nanoparticles tend to agglomerate upon storage for one week, as seen in Figure 3b,c.

X-ray diffraction (XRD) analysis confirms the amorphous nature of the synthesized bismuth nanoparticles (Figure 4).

This substantiates the diffraction pattern observed in Figure 1d, suggesting that the synthesized bismuth nanoparticles are amorphous in morphology due to the presence of phytochemicals enclosing these nanoparticles.

#### 2.2.2. Fourier Transform Infrared Spectroscopy (FT-IR) of the *M. oleifera* Leaves Extract and the Synthesized Bismuth Nanoparticles

The FT-IR spectra of the *M. oleifera* leaves extract and the synthesized bismuth nanoparticles are shown in Figure 5 and Figure 6, respectively. This vibration spectroscopy data can be used to identify the biomolecules implicated in the synthesis of bismuth nanoparticles. The bands around 3400 cm^−1^ and 1630 cm^−1^ are broad in the *M. oleifera* leaves extract (Figure 5), corresponding to the vibration mode of hydroxyl group, mostly found in polyphenolic molecules such as tannins, flavonoids, and glycoside derivatives. The FT-IR spectrum of the synthesized bismuth nanoparticles (Figure 6) depicts a sharp band at 3431 cm^−1^, corresponding to the N–H vibration mode. The fingerprint region at 1269 cm^−1^ explains the binding of the organic matrix to the synthesized bismuth nanoparticles. This comparison reflects the participation of the hydroxyl group in the synthesis of bismuth nanoparticles. Moreover, the binding characteristics of amino groups in the *M. oleifera* leaves extract are observed in Figure 5 and Figure 6. This could be the reason for agglomeration of the synthesized bismuth nanoparticles to give an amorphous morphology.

#### 2.2.3. UV-Vis Spectroscopy of the *M. oleifera* Leaves Extract and the Synthesized Bismuth Nanoparticles

The UV-Vis absorption spectrum of the *M. oleifera* leaves extract is shown in Figure 7, λ_max_ at 390 nm. The UV-Vis spectrum of the synthesized bismuth nanoparticles reconstituted in dimethyl sulfoxide (DMSO) solvent is shown in Figure 8, λ_max_ at 270 nm. The peak relative to the phytochemicals is slightly decreased, due to changes in poly-hydroxyl compounds responsible for bio-reduction and the encapsulation of the product, and the peak maximum is shifted to the right. The phytochemical complexed with bismuth are enriched in the Figure 8, and the absorbance at 220 and could be terpenes or polyphenolic species.

### 2.3. Antioxidant Activity of the M. oleifera Leaves Extract and the Synthesized Bismuth Nanoparticles

Using DPPH assay against ascorbic acid (standard), the antioxidant activity percentage (AA%) of the *M. oleifera* leaves extract and the synthesized bismuth nanoparticles was assessed. As shown in Table 3, the *M. oleifera* leaves extract exerted considerable antioxidant potential, while the synthesized bismuth nanoparticles displayed relatively weaker antioxidant activity. These results were supported with total antioxidant capacity measured by phosphomolybdate assay (Table 4).

### 2.4. Anti-Bacterial Activity of the M. oleifera Leaves Extract and the Synthesized Bismuth Nanoparticles

The potential anti-bacterial activity of synthesized bismuth nanoparticles was evaluated against four bacterial species (*E. coli*, *K. pneumoniae*, *S. aureus*, and *E. faecalis*). The classification of these species of bacteria and their localization in the human body is given in Table 5. The estimated minimum inhibitory concentration (MIC) values measured in presence of the *M. oleifera* leaves extract and the bismuth nanoparticles are given in Table 6. The growth of the indicated species of bacteria in the presence of different concentrations (7.8–1000 µg/mL) of *M. oleifera* leaves extract and the bismuth nanoparticles is shown in Table 7. The resazurin microtiter assay plates for the *M. oleifera* leaves extract and the synthesized bismuth nanoparticles are shown in Figure 9. In presence of the *M. oleifera* leaves extract, the estimated MIC values against *E. coli*, *K. pneumoniae*, *S. aureus*, and *E. faecalis* were found to be in the range of 250–500 µg/mL (Table 6). The estimated MIC values (250–500 µg/mL) in presence of the synthesized bismuth nanoparticles were similar to that of *M. oleifera* leaves extract (Table 6): the bismuth nanoparticles have comparable anti-bacterial activities (Table 6 and Figure 9), albeit a portion of the phytochemicals is complexed to the nanoparticles, as seen by the reduction in antioxidant power (Table 2). Hence, in the product bismuth nanoparticles, a reduction in TAC and AA% was observed with loss of some functional group during transformation of the phytochemicals, responsible for bio-reduction and the encapsulation of nanoparticles. This loss may account also to the difference in inhibition found, on *K. pneumoniae* and *S. aureus*, and on *Aspergillus* species.

### 2.5. Anti-Fungal Activity of the M. oleifera Leaves Extract and the Synthesized Bismuth Nanoparticles

The potential anti-fungal activity of synthesized bismuth nanoparticles was evaluated against four fungal species (*A. niger*, *A. flavus*, *C. albicans*, and *C. glabrata*). Ketoconazole (10 µg/500 µL) was used as a positive control, while water-ethanol (1:1) solution and DMSO solvent served as negative controls for the *M. oleifera* leaves extract and the bismuth nanoparticles, respectively. The Potato Dextrose Broth was also used as a negative control. The classification of these species of fungi and their localization in the human body is given in Table 8. The minimum inhibitory concentration (MIC) values measured in presence of the *M. oleifera* leaves extract and the bismuth nanoparticles are given in Table 6. The growth of the indicated species of fungi in presence of different concentrations (7.8–1000 µg/mL) of *M. oleifera* leaves extract and the bismuth nanoparticles is shown in Table 9. The resazurin microtiter assay plates for the *M. oleifera* leaves extract and the synthesized bismuth nanoparticles are shown in Figure 10. In presence of the *M. oleifera* leaves extract, the estimated MIC values against *A. niger*, *A. flavus*, *C. albicans,* and *C. glabrata* were found to be 62.5, 62.5, 125, and 250 µg/mL, respectively (Table 6). In presence of the synthesized bismuth nanoparticles, the estimated MIC values against *A. niger*, *A. flavus*, *C. albicans,* and *C. glabrata* were found to be 250, 250, 62.5, and 62.5 µg/mL, respectively (Table 6). These findings indicate that the synthesized bismuth nanoparticles displayed more effective anti-fungal activity against *C. albicans* and *C. glabrata* compared to the *M. oleifera* leaves extract (Table 6 and Table 9, and Figure 10). Candida species are normally present on human skin, and their infections are common in skin and other tissues due to dysbiosis and suppression of immune responses. Future in vivo analysis is required to shed light on the possible employment of the encapsulated phytochemical synthesized bismuth nanoparticles in the treatment of candidiasis, but we anticipate that they could serve as a promising therapeutic candidate to treat candidiasis after considering their pharmacokinetics.

## 3. Materials and Methods

### 3.1. Preparation of the M. oleifera Leaves Extract

The leaves of *M. oleifera* were collected from the wild shrub regions in Rajapalayam, Tamil Nadu, India. The plant was identified by the facility at Asthagiri Herbal Research Foundation, Chennai, India. The identified specimen vouchers (AHRF/HERBARIUM/021) were deposited at Asthagiri Herbal Research Foundation, Chennai, India. The country’s common breed was used in the study. The genetically-modified varieties of *M. oleifera* (PKM-1 and PKM-2) were not used in our study. Fresh *M. oleifera* leaves were collected and shade-dried. The dried leaves were smoothly crushed. The *M. oleifera* leaves powder was stored at room temperature. A total of 10 g of *M. oleifera* leaves powder was taken and soaked in 100 mL water-ethanol (1:1) solution. The mixture was macerated for 1 h and the *M. oleifera* leaves extract was filtered through Whatman filter paper 1. The *M. oleifera* leaves extract was used immediately for the preparation of bismuth nanoparticles and other experimental analyses.

### 3.2. Synthesis of the Bismuth Nanoparticles

Bismuth nitrate pentahydrate (1 g) was dissolved in 20 mL de-mineralized water and added to 80 mL of *M. oleifera* leaves extract. The reaction mixture was stirred for 3 h at 60 °C. The synthesized bismuth nanoparticles were collected by using centrifugation and repeatedly washed with de-mineralized water. The encapsulated phytochemical synthesized bismuth nanoparticles were dried at 105 °C for 1 h, and subsequently reconstituted in DMSO solvent.

### 3.3. Characterization of the Size and Morphology of the Synthesized Bismuth Nanoparticles

The completion of synthesis was characterized using a UV-3600 Plus UV-Vis double beam spectrophotometer (Shimadzu, Kyoto, Japan). The formation of bismuth nanoparticles and the role of biomolecules in this synthesis were confirmed using an ALPHA-E FTIR spectrometer (Bruker, Billerica, MA, USA). The crystalline nature of the synthesized bismuth nanoparticles was ascertained by XRD using a XRD-6000 diffractometer (Shimadzu, Kyoto, Japan). The size of the synthesized bismuth nanoparticles was measured by SEM using a FEI Quanta 200 scanning electron microscope, with the field emission gun (FEG) feature for better resolution (ThermoFisher Scientific, Waltham, MA, USA). The morphology assessment of the synthesized nanoparticles was performed by transmission electron microscopy (TEM) using a JEOL-TEM-2100 plus transmission electron microscopy (JEOL, Tokyo, Japan). The sample was dispersed in ethanol, coated on the grid, and dried for TEM analysis along with EDS analysis. In previous published work, several authors described improvement of silver nanoparticles, by means of a nitroxide coating to make silver NPs [43], through immobilization of TEMPO moieties on the surface of various materials and synthesis of novel hybrid nanostructures [44], and through synthesis of gold nanoparticles densely coated with nitroxide spins [45]. Although this is a preliminary work, in the next step we will perform thermogravimetric and XPS analyses to give a proper characterization of the synthesized materials and to assess the content of bismuth in the synthesized materials. We will deepen study of this area by applying this method in the future research. We are confident that the antifungal activity toward Candida species may provide a temporary therapeutic application.

### 3.4. Phytochemical Analysis of the M. oleifera Leaves Extract

The qualitative analysis of biomolecules present in the *M. oleifera* leaves extract was carried out for the presence of alkaloids, tannins, flavonoids, steroids, saponins, polyphenols, glycosides, carbohydrates, proteins, and amino acids. The total phenolic content in the *M. oleifera* leaves extract was estimated as gallic acid equivalent by Folin-Ciocalteu polyphenol assay [46]. The protein content in the *M. oleifera* leaves extract was estimated by using Lowry’s method [47].

### 3.5. Antioxidant Activity

#### 3.5.1. DPPH Assay (Antioxidant Activity Percentage—AA%)

The antioxidant activity percentage (AA%) (scavenging activity) of the *M. oleifera* leaves extract and the synthesized bismuth nanoparticles was assessed by using DPPH free radical scavenging assay. A total of 1 mg of ascorbic acid (standard) was dissolved in 1 mL of methanol. Different aliquots (serial dilution) of the ascorbic acid solution (0.1–0.5 mL), corresponding to 100–500 µg, were used for calibration. To each tube containing ascorbic acid solution, 1 mL of 0.1 mM DPPH radical solution in ethanol was added, and the final volume was adjusted to 4 mL using ethanol. The stock solutions for the *M. oleifera* leaves extract and the synthesized bismuth nanoparticles were prepared by dissolving 1 mg of each sample in 1 mL of an appropriate solvent (methanol for the *M. oleifera* leaves extract; DMSO for the synthesized bismuth nanoparticles). Different aliquots from the stock solutions (0.1–0.5 mL), corresponding to 100–500 µg, were added to all tubes except the blank tube (control). The volume in each tube was adjusted to 3 mL using ethanol. A total of 1 mL of 0.1 mM DPPH radical solution in ethanol was added to each tube,. The blank tube (control) was prepared by mixing 3 mL of ethanol and 1 mL of DPPH radical solution in ethanol. All tubes were incubated for 30 min at room temperature, and absorbance at 517 nm was recorded. AA% was determined using the following formula:
AA% = {(absorbance of blank) − (absorbance of sample)/(absorbance of blank)} × 100(1)


#### 3.5.2. Phosphomolybdenum Assay (Total Antioxidant Capacity–TAC)

The total antioxidant capacity (TAC) of the *M. oleifera* leaves extract and the synthesized bismuth nanoparticles was assessed by phosphomolybdenum assay as ascorbic acid equivalent. A total of 1 mg of ascorbic acid (standard) was dissolved in 1 mL of methanol. Different aliquots (serial dilution) of the ascorbic acid solution (0.1–0.5 mL), corresponding to 100–500 µg, were prepared. The volume in each tube was adjusted to 4 mL using distilled water. The stock solutions for the *M. oleifera* leaves extract and the synthesized bismuth nanoparticles were prepared by dissolving 1 mg of each sample in 1 mL of an appropriate solvent (methanol for the *M. oleifera* leaves extract; DMSO for the synthesized bismuth nanoparticles). Different aliquots from the stock solutions (0.1–0.5 mL), corresponding to 100–500 µg, were added to all tubes except the blank tube (control). The volume in each tube was adjusted to 3 mL using distilled water. To each tube, 1 mL of phosphomolybdenum reagent (0.6 M sulfuric acid, 28 mM sodium phosphate, and 4 mM ammonium molybdate) was added. The volume in each tube was adjusted to 4 mL using distilled water. After incubation for 90 min at 95 °C, absorbance at 695 nm was recorded. The calibration curve for the ascorbic acid solution (standard) was plotted for the absorbance at 695 nm against known amounts of ascorbic acid with the phosphomolybdate reagent in order to express the TAC values as ppm equivalent of ascorbic acid. Using the ascorbic acid calibration curve, the TAC values for the *M. oleifera* leaves extract and the synthesized bismuth nanoparticles were calculated and expressed as a ppm equivalent of ascorbic acid.

### 3.6. Anti-Bacterial Activity using Resazurin Microtiter Assay

The bacterial cultures of *E. coli*, *K. pneumoniae*, *S. aureus*, and *E. faecalis* were obtained from the Royal Bio Research Centre, Chennai, Tamil Nadu, India.The most rapid and inexpensive way to screen several microorganism isolates at the same time, with better correlation in comparison to other techniques, is the resazurin microtiter assay [48,49,50]. The resazurin solution was prepared by dissolving a 270 mg tablet of resazurin in 40 mL of sterile distilled water. The test was carried out in 96-well plates under aseptic conditions. A volume of 100 μL of sample containing 10 mg/mL was transferred into the first well of the plate. Afterwards, 50 μL of nutrient broth was added to all other wells, and the tested sample was serially diluted. Subsequently, 10 μL of resazurin solution and 10 μL of bacterial suspension were added to each well. To prevent dehydration, the plates were wrapped with cling film and incubated at 37 °C for 18–24 h. The color change was visually observed. A purple to pink (or colorless) color change was considered as being positive, indicating cell growth (i.e., (+) means growth and (-) means no growth). MIC was recorded at the lowest concentration whereby a color change occurred. Streptomycin (10 µg/500 µL) served as a positive control, while water-ethanol (1:1) solution and DMSO solvent were used as negative controls for the *M. oleifera* leaves extract and the bismuth nanoparticles, respectively. The nutrient broth was also used as a negative control.

### 3.7. Anti-Fungal Activity Using Resazurin Microtiter Assay

The fungal cultures of *A. niger*, *A. flavus*, *C. albicans*, and *C. glabrata* were obtained from the Royal Bio Research Centre, Chennai, Tamil Nadu, India.The resazurin microtiter assay was performed as described above (Section 3.6). Ketoconazole (10 µg/500 µL) was used as a positive control, while water-ethanol (1:1) solution and DMSO solvent served as negative controls for the *M. oleifera* leaves extract and the bismuth nanoparticles, respectively. The Potato Dextrose Broth was also used as a negative control.

## 4. Conclusion

Our experimental approach led to the successful green synthesis of bismuth nanoparticles using a hydroalcoholic extract of *M. oleifera* leaves. UV-Vis absorption and FT-IR spectrometry confirmed the formation of bismuth nanoparticles and the role of different biomolecules present in the *M. oleifera* leaves extract in the synthesis of bismuth nanoparticles. The presence of elemental bismuth was affirmed by EDS analysis. XRD analysis confirmed the crystalline nature of the synthesized bismuth nanoparticles. SEM analysis revealed that the size of the synthesized bismuth nanoparticles is in the range of 40.4–57.8 nm. As revealed by TEM with EDS, the synthesized bismuth nanoparticles were confirmed to display an amorphous morphology. DPPH and phosphomolybdenum assays revealed that both the *M. oleifera* leaves extract and the synthesized bismuth nanoparticles exert considerable antioxidant activity. However, the antioxidant potential of the synthesized bismuth nanoparticles is weaker than that of the *M. oleifera* leaves extract, most likely due to a loss of some phenolic constituents. Our study further demonstrated that the *M. oleifera* leaves extract and the synthesized bismuth nanoparticles play a potent anti-bacterial role against *E. coli*, *K. pneumoniae*, *S. aureus,* and *E. faecalis* (MIC values for the extract: 500, 250, 250, and 250 µg/mL; MIC values for the bismuth nanoparticles: 500, 500, 500, and 250 µg/mL, respectively), with a more profound inhibitory effect of the bismuth nanoparticles, compared to the *M. oleifera* leavesextract, against *E. faecalis*. Moreover, our findings also revealed that the *M. oleifera* leaves extract and the synthesized bismuth nanoparticles display relatively stronger anti-fungal activity against *A. niger*, *A. flavus*, *C. albicans,* and *C. glabrata* (MIC values for the extract: 62.5, 62.5, 125, and 250 µg/mL; MIC values for the bismuth nanoparticles: 250, 250, 62.5, and 62.5 µg/mL, respectively), with a promising and exploitable inhibitory activity against Candida species. This indicates that the anti-fungal effects of the *M. oleifera* leaves extract against *A. niger* and *A. flavus* is stronger than that of the synthesized bismuth nanoparticles, while the latter exerts more potent anti-fungal effects on *C. albicans* and *C. glabrata*. Obviously, our green synthesis approach to obtain bismuth nanoparticles using a hydroalcoholic extract of *M. oleifera* leaves did not jeopardize the bismuth nanoparticles’ anti-bacterial and anti-fungal activities. Collectively, our study implies that the synthesized bismuth nanoparticles can be potentially used in the amelioration of oxidative stress and various microbial infections. In particular, the synthesized bismuth nanoparticles seem to be powerful anti-fungal agents against cases of candidiasis including, but not limited to oral thrush, urinary yeast infection, genital infection, intra-abdominal candidiasis, acute hematogenous candidiasis, fungemia, invasive candidiasis, endocarditis, and endophthalmitis. One advantage of green synthesis of nanoparticles is that it yields non-toxic products and it reduces the production of wasteful material. Another advantage of green synthesis of nanoparticles is that the encapsulating material being polyphenols has a therapeutic value compared to the incinerated nanoparticles that are synthesized by using conventional thermal-chemical methods.

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
