# Peer review of "Use of A Hydroalcoholic Extract of Moringa oleifera Leaves for the Green Synthesis of Bismuth Nanoparticles and Evaluation of Their Anti-Microbial and Antioxidant Activities"

_materials, 2020, doi:10.3390/ma13040876_

Round 1

Reviewer 1 Report

This is a quite well conducted study, with a few methodological issues in the bacteriological assessment part:

3.2 Please refer to the bismuth nanoparticles as bismoth-MO nanopaticles (combined wit M. oleifera).

3.6 Please mention the concentration of bacterial inoculum

3.7 The antifungal activity should have been performed in RPMI-1640 with 2% glucose medium, not nutrient broth. Tween may be added for homogenous spore distribution in the Asperfillus spp. inoculum. Please repeat if possible in these conditions, otherwise either lose this small chapter or mention as a limitation (the detected MICs are not real, but comparable between the samples)

Tables 5 and 8 - placed in the methods section, they are not results.

Table 5 - check the spelling: upper respiratory tract; Klebsiella located in the upper respiratory tract (not mouth, not skin).

Tables 7 and 10 are not required, can be deducted from the figures.

Table 8 - Aspergillus spp. are not part of the commensal microflora; not at all in the lungs! May produce infections in the reported sites. C. albicans - also in mucosal tissues.

Figure 9 - note the columns for the controls

Figure 10-b is distorted

Other issues:

rows 205-209 and 227-231- should be deleted, they are mentioned in the methods section

rows 242-244 -  the highest prevalence of candidiasis is mainly after dysbiosis, secondly in immune-depression.

Discussions could be more consistent and the results can be reported to other publications on the topic. Too many references were condensed in the introduction. The fist 2/3 of the conclusions can be reported to the literature.

371-373 - the clinical use cannot be speculated without some in-vivo studies!

Author Response

Reviewer #1:

Comments and Suggestions for Authors

This is a quite well conducted study, with a few methodological issues in the bacteriological assessment part:

3.2 Please refer to the bismuth nanoparticles as bismuth-MO nanoparticles (combined with M. oleifera).

The product is phytochemical encapsulated bismuth nanoparticles

3.6 Please mention the concentration of bacterial inoculum.

A loop full of bacterial culture was inoculated.

3.7 The antifungal activity should have been performed in RPMI-1640 with 2% glucose medium, not nutrient broth. Tween may be added for homogenous spore distribution in the Asperfillus spp. inoculum. Please repeat if possible in these conditions, otherwise either lose this small chapter or mention as a limitation (the detected MICs are not real, but comparable between the samples)

Potato Dextrose Broth is used for the anti-fungal activity studies.

Tables 5 and 8 - placed in the methods section, they are not results.

These tables were placed in the discussion section to have a better understanding of the presented results.

Table 5 - check the spelling: upper respiratory tract; Klebsiella located in the upper respiratory tract (not mouth, not skin).

We agree with the reviewer. Both mistakes were corrected in the revised manuscript (RED).

Tables 7 and 10 are not required, can be deducted from the figures.

Table 7 and Table 10 give complete data on the sample, positive control, negative control and nutrient broth in the antimicrobial studies. So, we strongly believe that their presence is relevant.

Table 8 - Aspergillus spp. are not part of the commensal microflora; not at all in the lungs! May produce infections in the reported sites. C. albicans - also in mucosal tissues.

We agree with the reviewer. The needed corrections were made accordingly in Table 5 and Table 8.

Figure 9 - note the columns for the controls

In Figure 9, the control is noted as “std” – meaning standard (in this case, streptomycin is the control). Likewise, in Figure 10, the control is noted as “std” – meaning standard (in this case, ketoconazole is the control).

Figure 10 is distorted

Figure 10 was rectified in the revised manuscript.

Other issues:

Rows 205-209 and 227-231- should be deleted, they are mentioned in the methods section.

As suggested by the reviewer, the indicated duplicated sentences were removed in the revised manuscript.

Rows 242-244 - the highest prevalence of candidiasis is mainly after dysbiosis, secondly in immune-depression.

As suggested by the reviewer, we changed the corresponding sentence in the revised manuscript, which now reads as “Candida species are normally present on human skin, and their infections are common in skin and other tissues due to dysbiosis and suppression of immune responses”.

Discussions could be more consistent and the results can be reported to other publications on the topic. Too many references were condensed in the introduction. The first 2/3 of the conclusions can be reported to the literature.

The revisions in the revised manuscript reflect consistency of the reported results and their discussions. The given references provide a coherent background to the present study and they support the reported findings of the present study and previous research work.

371-373 - the clinical use cannot be speculated without some in-vivo studies!

We totally agree with the reviewer. The sentence was changed in the revised manuscript to read as follows (RED): “Future in vivo analysis is required to shed light on the possible employment of the phytochemical encapsulated synthesized bismuth nanoparticles in the treatment of candidiasis, but we anticipate that they could serve as a promising therapeutic candidate to treat candidiasis after considering their pharmacokinetics”.

Reviewer 2 Report

The author used a hydroalcoholic extract of  Moringa oleifra leave to synthesize bismuth nanoparicles and its antimicrobial activity. 

However, the data and the way to perform needed to be improved a lot. sloppy.several questions needed to be answered:

1. bismuth nanoparicles need a XPS data to show it is Bi nanoparticles and also it is better to have Bi standard compound to compare.

2. what kind of medium did you grow your bacteria, and what are strain's number?? there is no document in this paper. you have to follow the standard MIC protocol, growing in MHB medium.

3. there is no control data for MIC, and do you also test normal Bi , not the Bi nanoparticles to compare your MIC?? need to add this data. I donot know what is novel? Bi nanoparticles is better than BI for antimicrobial activity?

4. deleted figure 9, and 10 no needed

5. combine table 5,6, 7 there are the same stuff, put in a table.

6. combine table 8,9, 10 there are the same stuff, put in a table.

7. what is different figure 5, and 6, what does these two means??does it is important ?

does it related to Bi nanoparticles? why do you need these two figure???

Author Response

Reviewer #2:

Comments and Suggestions for Authors

The author used a hydroalcoholic extract of Moringa oleifera leave to synthesize bismuth nanoparticles and its antimicrobial activity.

However, the data and the way to perform needed to be improved a lot. sloppy. several questions needed to be answered:

bismuth nanoparicles need a XPS data to show it is Bi nanoparticles and also it is better to have Bi standard compound to compare.

X-ray energies for bismuth Mα and Lα lines at 2.469 KeV and 10.839 KeV proves the presence of elemental bismuth in the encapsulated nanoparticles.

what kind of medium did you grow your bacteria, and what are strain's number?? there is no document in this paper. you have to follow the standard MIC protocol, growing in MHB medium.

Resazurin Microtiter Assay for MIC protocol was followed.  The stains were cultured and authenticated by Royal BioResearch Centre, Chennai, India.

there is no control data for MIC, and do you also test normal Bi , not the Bi nanoparticles to compare your MIC?? need to add this data. I donot know what is novel? Bi nanoparticles is better than BI for antimicrobial activity?

Streptomycin (control for anti-bacterial activity) and Ketoconazole (control for anti-fungal activity).

Novelty – the hydroalcoholic extract of M. oleifera leaves was used to synthesize bismuth nanoparticles by green method, resulting in phytochemical encapsulated bismuth nanoparticles with remarkably good anti-bacterial and anti-fungal activity. One advantage of green synthesis of nanoparticles is that the encapsulating material being polyphenols has a therapeutic value compared to the incinerated nanoparticles that are synthesized by conventional thermal-chemical methods.

deleted figure 9, and 10 no needed

Resazurin microtiter assay plates images give a better understanding of the data tabulated.

combine table 5, 6, 7 there are the same stuff, put in a table.

Table 5, 6, and 7 have its own significance, so keeping them separate give better clarity to the results.

combine table 8,9, 10 there are the same stuff, put in a table.

Table 8, 9, and 10 have their own significance, so keeping them separate give better clarity to the results.

what is different figure 5, and 6, what does these two means??does it is important ?

does it related to Bi nanoparticles? why do you need these two figure???

Figure 5 and 6 are FT-IR spectra of the hydroalcoholic extract of M. oleifera leaves and the synthesized bismuth nanoparticles. The vibrational spectral comparison shows the participation of polyphenolic chemical constituents in the M. oleifera leaves extract towards the bio-reduction, followed by stabilization of the bismuth nanoparticles through encapsulation.

Reviewer 3 Report

The manuscript entitled "" reports the sinthesis of bismuth nanoparticles using a hydroalcoholic extract of M. oleifera leaves and demonstrates their potential synergy benefits. Overall the manuscript is well written and organized. The information is clear and the data is pertinent. However, the discussion of this work is very poor... the authors basically report the results and present the conclusions but barely discuss the results and the implications they may have. The discussion should definetly be improved and a critical analysis must join the results and discussion section, with the comparision of the data with previously published works.

Please reduce the abstract, this is not suppose to give all the details only an overview of the entire research. There are too many details here that should be eliminated.

There is a lot of agglomeration between the nanoparticles. How do you intend to overcome this situation?

Author Response

Reviewer #3:

Comments and Suggestions for Authors

The manuscript entitled "" reports the synthesis of bismuth nanoparticles using a hydroalcoholic extract of M. oleifera leaves and demonstrates their potential synergy benefits. Overall the manuscript is well written and organized. The information is clear and the data is pertinent. However, the discussion of this work is very poor... the authors basically report the results and present the conclusions but barely discuss the results and the implications they may have. The discussion should definetly be improved and a critical analysis must join the results and discussion section, with the comparision of the data with previously published works.

We thank the reviewer for this comment. The presented results are discussed and deliberated as much as possible. We further revised the discussion to shed more light on the meaning of the results.

Please reduce the abstract, this is not suppose to give all the details only an overview of the entire research. There are too many details here that should be eliminated.

We agree with the reviewer. The abstract was reduced from 377 to 294 words by eliminating details that are mentioned in other sections of the manuscript.

There is a lot of agglomeration between the nanoparticles. How do you intend to overcome this situation?

The encapsulation of bismuth nanoparticles by the phytochemical constituents in the M. oleifera leaves extract can extend to agglomeration upon storage (standing in reaction medium). During the synthesis, the bismuth nanoparticles were washed with water to remove the extract, but simple washing doesn’t completely dissociate the binding of macromolecules that had encapsulated the nanomaterial. To remove these phytochemicals, harsh chemical and thermal treatments are required, which were never adopted in this study in order to present a simple green synthetic product. One advantage of green synthesis of nanoparticles is that the encapsulating material being polyphenols has a therapeutic value compared to the incinerated nanoparticles that are synthesized by conventional thermal-chemical methods.

Reviewer 4 Report

Recommendation: I recommend the manuscript “Use of a hydroalcoholic extract of Moringa oleifera leaves for the green synthesis of bismuth nanoparticles and evaluation of their anti-microbial and antioxidant activities” by Prince Edwin Das, Amin F. Majdalawieh, Imad A. Abu-Yousef, Srinivasan Narasimhan, and Palmiro Poltronieri for publication in the Materials journal.

The manuscript provides valuable insights on the nanomaterial prepared using biocompatible and bioactive reagents. The prepared in this way bismuth nanoparticles may find interesting applications as it was showed in the manuscript they exhibit both biocidal and antioxidant activity.

However, in my opinion the synthesized material is not well-defined. For example I am not convinced on the basis of the results presented in the manuscript that the presented results were obtained for nanoparticles with size ca 40 nm or their agglomerates/aggregates. I think that several studies/measurements should be added. Furthermore, the manuscript should in some places be improved.

Comments:

There is a lack of direct proof on the presence of bismuth nanoparticles inside of the prepared materials. In the presented XRD patterns the diffraction peaks assigned to bismuth nanoparticles are not clearly visible. In TEM images presented in Fig. 1 nanoparticles are not visible, a huge amount of organic matter makes it impossible to see whether we have there nanoparticles or their aggregates. I think that the prepared material should be first purified from the excess of organic material. I recommend a dialysis for this purpose. So, in summary, the TEM analysis should be performed for purified material and then could be presented in the manuscript. The presented SEM image cannot be used to determine size and shape of nanoparticles, like the Authors have done this, absolutely. It a lack of information about content of bismuth in the synthesized materials. It is a serious drawback. Thermogravimetric or XPS analyses should performed to give a proper characterization of the synthesized materials (see RSC Advances, 5(2015), 58403-58415, these references should be added) Antibacterial activity of nanoparticles depends directly on their size, shape and surface coating. Therefore, the nanoparticles should be characterized in details in this regard. Actually, in the presented TEM images the bismuth nanoparticles are not visible. The obtained MICs are very high, antibacterial activity of these materials very weak. Probably, the nanoparticles are aggregated in the prepared materials. It should be explained by the Authors. The presented in the manuscript UV-vis spectra should be discussed in more details. It is a crucial issue when we want to characterize nanoparticles with SPR. For bismuth nanoparticles SPR band’s maximum should be located at 265 nm, in Fig. 8 we can see a peak with maximum 270 nm why it is shifted? Please explain in a view of size and shape of the synthesized nanoparticles. It is not clear how the total phenolic content was determined. Authors gave an information that it was estimated as an equivalent of gallic acid by Folin-Ciocalteu polyphenol assay with the reference [43] but I think that general information on kind of measurements should be added. Several references should be added:RSC Advances, 2015, 5, 58403 – 58415.

Author Response

Comments and Suggestions for Authors

Recommendation: I recommend the manuscript “Use of a hydroalcoholic extract of Moringa oleifera leaves for the green synthesis of bismuth nanoparticles and evaluation of their anti-microbial and antioxidant activities” by Prince Edwin Das, Amin F. Majdalawieh, Imad A. Abu-Yousef, Srinivasan Narasimhan, and Palmiro Poltronieri for publication in the Materials journal.

The manuscript provides valuable insights on the nanomaterial prepared using biocompatible and bioactive reagents. The prepared in this way bismuth nanoparticles may find interesting applications as it was showed in the manuscript they exhibit both biocidal and antioxidant activity.

However, in my opinion the synthesized material is not well-defined. For example I am not convinced on the basis of the results presented in the manuscript that the presented results were obtained for nanoparticles with size ca 40 nm or their agglomerates/aggregates. I think that several studies/measurements should be added. Furthermore, the manuscript should in some places be improved.

The manuscript was revised accordingly, taken the reviewer’s observation into consideration.

Comments:

There is a lack of direct proof on the presence of bismuth nanoparticles inside of the prepared materials. In the presented XRD patterns the diffraction peaks assigned to bismuth nanoparticles are not clearly visible. In TEM images presented in Fig. 1 nanoparticles are not visible, a huge amount of organic matter makes it impossible to see whether we have there nanoparticles or their aggregates. I think that the prepared material should be first purified from the excess of organic material. I recommend a dialysis for this purpose. So, in summary, the TEM analysis should be performed for purified material and then could be presented in the manuscript. The presented SEM image cannot be used to determine size and shape of nanoparticles, like the Authors have done this, absolutely. It a lack of information about content of bismuth in the synthesized materials. It is a serious drawback. Thermogravimetric or XPS analyses should performed to give a proper characterization of the synthesized materials (see RSC Advances, 5(2015), 58403-58415, these references should be added) Antibacterial activity of nanoparticles depends directly on their size, shape and surface coating. Therefore, the nanoparticles should be characterized in details in this regard. Actually, in the presented TEM images the bismuth nanoparticles are not visible. The obtained MICs are very high, antibacterial activity of these materials very weak. Probably, the nanoparticles are aggregated in the prepared materials. It should be explained by the Authors. The presented in the manuscript UV-vis spectra should be discussed in more details. It is a crucial issue when we want to characterize nanoparticles with SPR. For bismuth nanoparticles SPR band’s maximum should be located at 265 nm, in Fig. 8 we can see a peak with maximum 270 nm why it is shifted? Please explain in a view of size and shape of the synthesized nanoparticles. It is not clear how the total phenolic content was determined. Authors gave an information that it was estimated as an equivalent of gallic acid by Folin-Ciocalteu polyphenol assay with the reference [43] but I think that general information on kind of measurements should be added. Several references should be added: RSC Advances, 2015, 5, 58403 – 58415.

The nanoparticles synthesized using plant extract will be agglomerated due to the macromolecules in the extract. This encapsulation can be removed if the final product is heat-treated at 500-700°C, wherein the organic moieties will be evaporated to give ashes. So, in this study, the encapsulated bismuth nanoparticles were used as such for physical characterization and assessment of antimicrobial activity. Being green synthesis, an attempt to convert the product to ash by roasting was completely avoided. One advantage of green synthesis of nanoparticles is that the encapsulating material being polyphenols has a therapeutic value compared to the incinerated nanoparticles that are synthesized by conventional thermal-chemical methods.

The phytochemical encapsulated bismuth nanoparticles were freshly prepared. The size and morphology of the bismuth nanoparticles were studied with this sample.  The HRTEM and SEM images captured for the sample were updated in the revised manuscript. Likewise, the XRD image for the encapsulated bismuth nanoparticles were updated in the revised manuscript.

Round 2

Reviewer 2 Report

The authors did not answer the reviewers questions 

1.combine the table 6, 9 as a table  it is redundancy to put into two tables.

2.figure 9, and 10 , the control color is red, how do you measure at OD 600 nm , it is not good measurement  for bacteria.

can you explain it , how do you explain it black color and red color can be measure at OD 600 nm? which reference?

I ask you to double check with CFU/ml with your MIC , I donot think it is a c The authors did not answer the reviewers questions 

1.combine the table 6, 9 as a table  it is redundancy to put into two tables.

2.figure 9, and 10 , the control color is red, how do you measure at OD 600 nm , it is not good measurement  for bacteria.

can you explain it , how do you explain it black color and red color can be measure at OD 600 nm? which reference?

I ask you to double check with CFU/ml with your MIC , I do not think it is a correct way to represent MIC.

your background color is too red and dark correct way to represent MIC.

your background color is too red and dark...

Author Response

combine the table 6, 9 as a table it is redundancy to put into two tables.

Table 6a and Table 6b (previously 9) provide estimated Minimum Inhibitory Concentration (MIC) values for anti-bacterial and anti-fungal activity, respectively. Both tables were combined into 6a and 6b.

figure 9, and 10 , the control color is red, how do you measure at OD 600 nm , it is not good measurement for bacteria.

Sorry for misunderstanding: No OD values were used. As mentioned in section 3.6 (Materials and Methods), the color change was visually observed in the plates by means of resazurin assay (growth/no growth).

I ask you to double check with CFU/ml with your MIC, I do not think it is a correct way to represent MIC. Thank you for your comment. You are right to specifiy this is not minimum inhibitory activity.

MIC values were only estimated, based on the color change, which was visually observed in the plates (growth/no growth). See refs 48-50.

In the revised manuscript, this is clearly mentioned in section 3.6 (Materials and Methods). We want to highlight that the purpose of these experiments was to check the potential anti-bacterial and anti-fungal activities of the phytochemical-encapsulated bismuth nanomaterial. Hence, CFU/mL measurements were not required.  Sorry for the misuse of the word MIC. The word “estimated” was added wherever MIC is mentioned throughout the revised manuscript.

can you explain it , how do you explain it black color and red color can be measure at OD 600 nm? which reference?

OD measurement at 600 nm was not part of this study. c

your background color is too red and dark.

Thank you for your comments. The new figures are more lighter. As mentioned in section 3.6 (Materials and Methods), the color change was visually observed in the plates. The background color and the color in all wells was not experimentally difficult to recognize/distinguish. 

Reviewer 3 Report

Even though the discussion is still a bit superficial, the authors did a good job responding to the reviewers' comments, and updating and altering the information accordingly. I now recommend this manuscript publication.

Author Response

We thank the reviewer for acknowledging our revision and adequately responding to his/her comments. We thank the reviewer for recommending publication of our manuscript. We have improved a little bit more the revised manuscript.

Reviewer 4 Report

I see that the Authors ignored my comments and the last submitted version doesn't differ significant form the previous version of the manuscript, so I can only give again the same comments to their manuscipts as below

Recommendation: I recommend the manuscript “Use of a hydroalcoholic extract of Moringa oleifera leaves for the green synthesis of bismuth nanoparticles and evaluation of their anti-microbial and antioxidant activities” by Prince Edwin Das, Amin F. Majdalawieh, Imad A. Abu-Yousef, Srinivasan Narasimhan, and Palmiro Poltronieri for publication in the Materials journal.

The manuscript provides valuable insights on the nanomaterial prepared using biocompatible and bioactive reagents. The prepared in this way bismuth nanoparticles may find interesting applications as it was showed in the manuscript they exhibit both biocidal and antioxidant activity.

However, in my opinion the synthesized material is not well-defined. For example I am not convinced on the basis of the results presented in the manuscript that the presented results were obtained for nanoparticles with size ca 40 nm or their agglomerates/aggregates. I think that several studies/measurements should be added. Furthermore, the manuscript should in some places be improved.

Comments:

There is a lack of direct proof on the presence of bismuth nanoparticles inside of the prepared materials. In the presented XRD patterns the diffraction peaks assigned to bismuth nanoparticles are not clearly visible. In TEM images presented in Fig. 1 nanoparticles are not visible, a huge amount of organic matter makes it impossible to see whether we have there nanoparticles or their aggregates. I think that the prepared material should be first purified from the excess of organic material. I recommend a dialysis for this purpose.

So, in summary, the TEM analysis should be performed for purified material and then could be presented in the manuscript. The presented SEM image cannot be used to determine size and shape of nanoparticles, like the Authors have done this, absolutely.

It a lack of information about content of bismuth in the synthesized materials. It is a serious drawback. Thermogravimetric or XPS analyses should performed to give a proper characterization of the synthesized materials (see RSC Advances, 5(2015), 58403-58415; Polyhedron (2012) 46, 119-123, these references should be added) Antibacterial activity of nanoparticles depends directly on their size, shape and surface coating. Therefore, the nanoparticles should be characterized in details in this regard. Actually, in the presented TEM images the bismuth nanoparticles are not visible. The obtained MICs are very high, antibacterial activity of these materials very weak. Probably, the nanoparticles are aggregated in the prepared materials. It should be explained by the Authors. The presented in the manuscript UV-vis spectra should be discussed in more details. It is a crucial issue when we want to characterize nanoparticles with SPR. For bismuth nanoparticles SPR band’s maximum should be located at 265 nm, in Fig. 8 we can see a peak with maximum 270 nm why it is shifted? Please explain in a view of size and shape of the synthesized nanoparticles. It is not clear how the total phenolic content was determined. Authors gave an information that it was estimated as an equivalent of gallic acid by Folin-Ciocalteu polyphenol assay with the reference [43] but I think that general information on kind of measurements should be added. Several references should be added:

Advances in Colloid and Interface Science 250 C (2017) pp. 158-184;

RSC Advances, 2015, 5, 58403 – 58415

Author Response

So, in summary, the TEM analysis should be performed for purified material and then could be presented in the manuscript. The presented SEM image cannot be used to determine size and shape of nanoparticles, like the Authors have done this, absolutely.

Thank you so much for this insightful comment. We Wrote:

187-189: SEM images suggest an approximate size of the synthesized bismuth particles in the range of 40 and 60 nm. It is envisaged to perform a TEM analysis to obtain proper size of nanoparticles. However, SEM analysis suggests that phytochemical encapsulated bismuth nanoparticles tend to agglomerate upon storage for one week, as seen in Figure 3(b) and Figure 3(c).

There is a lack of information about content of bismuth in the synthesized materials. It is a serious drawback. Thermogravimetric or XPS analyses should be performed to give a proper characterization of the synthesized materials (see RSC Advances, 5(2015), 58403-58415; Polyhedron (2012) 46, 119-123, these references should be added) Antibacterial activity of nanoparticles depends directly on their size, shape and surface coating. Therefore, the nanoparticles should be characterized in details in this regard. Actually, in the presented TEM images the bismuth nanoparticles are not visible.

A: Thank you for this comment. WE appreciate your supervision. We wrote: 350-355 Albeit this is a preliminary work, in the next step we will perform thermogravimetric and XPS analyses to give a proper characterization of the synthesized materials and to assess the content of bismuth in the synthesized materials. We will deepen the study applying this method in the future research. We are confident that the antifungal activity toward Candida species may provide a temporary therapeutic application.

The obtained MICs are very high, antibacterial activity of these materials very weak. Probably, the nanoparticles are aggregated in the prepared materials. It should be explained by the Authors. The presented in the manuscript UV-vis spectra should be discussed in more details. It is a crucial issue when we want to characterize nanoparticles with SPR. For bismuth nanoparticles SPR band’s maximum should be located at 265 nm, in Fig. 8 we can see a peak with maximum 270 nm: why it is shifted? Please explain in a view of size and shape of the synthesized nanoparticles.

A: Due to the presence of organic matrix engulfing the nanomaterial, the absorption wavelength maxima was at 270 nm, shifting the one at 265 nm, which is the SPR band. Surface chemistry of the particle is one of the reasons for red-shift.  

UV-Vis spectrum of the synthesized bismuth nanoparticles reconstituted in dimethyl sulfoxide (DMSO) solvent is shown in Figure 8, λmax at 270 nm. The peak relative to the phytochemicals is slightly decreased, due to changes in poly-hydroxyl compounds responsible for bio-reduction and the encapsulation of the product, and the peak maximum is shifted to the right. The phytochemical complexed with bismuth are enriched in the Fig. 8, and the absorbance at 220 and could be terpenes or polyphenolic species.

It is not clear how the total phenolic content was determined. Authors gave an information that it was estimated as an equivalent of gallic acid by Folin-Ciocalteu polyphenol assay with the reference [43] but I think that general information on kind of measurements should be added. Several references should be added, as Advances in Colloid and Interface Science 250 C (2017) pp. 158-184;

Author: Thank you.

The scope of the study is to synthesize bismuth nanomaterial using plant extract through green synthesis method. This would result in phytochemical encapsulated bismuth nanomaterial. So, the organic matrix seen in HRTEM and SEM images were the encapsulation from the phytochemicals present in thehydroalcoholicextract of oleifera leaves. Hence, for characterization and anti-bacterial/anti-fungal studies, the material was washed, vortexed, and centrifuged (several times) with water and ethanol to remove any excess/leftover of the extract. So that dialysis was not required. As such, we need the nanomaterial in its phytochemical-encapsulated form for experimentation. Energy dispersive x-ray spectroscopy (EDS):the counts at 2.469 KeV and 10.839 KeV correspond to the X-ray energies for bismuth Mα and Lα lines. This is the direct evidence as the probe is focused on the region in Figure 1c. X-ray photoelectron spectroscopy (XPS) would provide similar information. Hence,there was no need to perform XPS. The work from the scope of green synthesis was extended to observe the anti-bacterial/anti-fungal activity of the phytochemical-encapsulated bismuth nanomaterial. This is to appreciate the viability of the product applications against pathogenic organisms. The presence of organic matrix encapsulating/engulfing the nanomaterial, the absorption wavelength maxima was at 270 nm and not at 265 nm, which is a Surface Plasmon Resonance (SPR). Surface chemistry of the particle is one of the reasons for a “red-shift”. So, the data was presented as such.

Author:

The phytochemicals present in M. oleifera extract were responsible for the AA% and TAC.  It is these poly-hydroxylcompounds responsible for bio-reduction and the encapsulation of the product.  Hence, in the product bismuth nanoparticles, there will be reduction in TAC and AA% due to the absence/functional group transformation of these phytochemicals.

The total polyphenol content was assessed by Folin-Ciocalteu polyphenol assay before and after the synthesis of copper nanomaterial.  The difference of 60 mg of gallic acid equivalent per 10g of the dried leaf was observed. Also, from 0.04M bismuth nitrate solution, we were able to synthesize 170 mg of bismuth nanomaterial. Folin-Ciocalteu polyphenol assay was performed as per the given reference.

As suggested by the reviewer, we added the indicated reference (Megiel, E. Surface modification using TEMPO and its derivatives. Colloid Interface Sci. 2017, 250, 158-184),  the reference RSC Advances, 5(2015), 58403-58415 now #43, and the other one, Polyhedron 2012

Round 3

Reviewer 2 Report

need to redo All your MIC, the purple and black color is not suitable for OD 600 nm detection, not for visible light.

You need to do plate cell number counting ( coloning forming unit) to  check your MIC, I donot think this is a correct way to do science. the author's answer , did not answer this question and try to excuse it. 

If you are NOT answer me the MIC problem, I will reject it because this is a wrong way to do science.the purple and black color in your plate  is not suitable for OD 600 nm detection, not for visible light.

Reviewer 4 Report

The Authors have introduced in the revised version of the manuscript some changes, they gave some explanations.

In my opinion the manuscript is now ready to be published in the Materials journal.